# EWS/FLI1 Characterization, Activation, Repression, Target Genes and Therapeutic Opportunities in Ewing Sarcoma

**DOI:** 10.3390/ijms242015173

**Published:** 2023-10-14

**Authors:** Muhammad Yasir, Jinyoung Park, Wanjoo Chun

**Affiliations:** Department of Pharmacology, Kangwon National University School of Medicine, Chuncheon 24341, Republic of Korea; yasir.khokhar1999@gmail.com (M.Y.); jinyoung0326@kangwon.ac.kr (J.P.)

**Keywords:** EWS/FLI1, therapeutic opportunities, Ewing sarcoma target genes

## Abstract

Despite their clonal origins, tumors eventually develop into complex communities made up of phenotypically different cell subpopulations, according to mounting evidence. Tumor cell-intrinsic programming and signals from geographically and temporally changing microenvironments both contribute to this variability. Furthermore, the mutational load is typically lacking in childhood malignancies of adult cancers, and they still exhibit high cellular heterogeneity levels largely mediated by epigenetic mechanisms. Ewing sarcomas represent highly aggressive malignancies affecting both bone and soft tissue, primarily afflicting adolescents. Unfortunately, the outlook for patients facing relapsed or metastatic disease is grim. These tumors are primarily fueled by a distinctive fusion event involving an FET protein and an ETS family transcription factor, with the most prevalent fusion being EWS/FLI1. Despite originating from a common driver mutation, Ewing sarcoma cells display significant variations in transcriptional activity, both within and among tumors. Recent research has pinpointed distinct fusion protein activities as a principal source of this heterogeneity, resulting in markedly diverse cellular phenotypes. In this review, we aim to characterize the role of the EWS/FLI fusion protein in Ewing sarcoma by exploring its general mechanism of activation and elucidating its implications for tumor heterogeneity. Additionally, we delve into potential therapeutic opportunities to target this aberrant fusion protein in the context of Ewing sarcoma treatment.

## 1. Introduction

Ewing sarcoma is an infrequent tumor that predominantly develops in the bones of children and adolescents [1]. Although there have been notable advancements in treatment over recent decades, survival rates have remained disappointingly low, even among patients with localized Ewing sarcoma. This is due to a significant portion of these tumors showing resistance to standard therapies, leading to frequent relapses. Furthermore, approximately a quarter of cases are diagnosed with disseminated disease, which carries an exceedingly grim prognosis [2]. Hence, there is a pressing demand for novel targeted therapies that can provide improved efficacy and minimize adverse effects compared to the conventional chemotherapy and radiotherapy currently in use.

In this regard, gaining insights into the molecular mechanisms underlying Ewing sarcoma pathogenesis offers valuable knowledge that could aid in the development of innovative targeted biological treatments. Ewing sarcomas are distinguished by chromosomal translocations that merge the EWSR1 gene with certain members of the ETS family of transcription factors, with FLI1 being the most commonly implicated [t(11;22) (q24;q12)] [3,4]. The EWS/FLI1 fusion protein functions as an abnormal transcription factor crucial for the development of Ewing tumors. It plays a pivotal role in controlling the expression of numerous target genes and orchestrating the oncogenic processes responsible for the malignant transformation of precursor cells into cancerous ones. Considering that the oncogenic attributes of EWS/FLI1 hinge on its ability to activate or suppress particular target genes [5], these target genes also present intriguing prospects for the discovery of novel targeted therapeutic approaches.

Over the recent years, substantial endeavors have been dedicated to uncovering the functional significance of genes regulated by EWS/FLI1 in the context of Ewing sarcoma pathogenesis. Consequently, numerous genes that hold key roles in Ewing sarcoma have been elucidated [5,6,7]. This effort has unveiled crucial molecular pathways implicated in Ewing sarcoma pathogenesis, offering novel molecular targets. An exhaustive examination of all the EWS/FLI1 target genes identified thus far and their relevance in targeted therapy exceeds the scope of this review. Therefore, our focus here is on a curated selection of seven EWS/FLI1 target genes, which we believe hold promise for future investigations and the potential discovery of fresh therapeutic strategies.

## 2. Characterization of EWS/FLI1

EWS/FLI1 plays a pivotal role in promoting tumorigenesis, cell proliferation, and the expansion of tumors [8]. Nonetheless, the transformation and tumorigenicity driven by EWS/FLI1 hinge on the expression level and activity of this fusion oncogene. In cell types that are not conducive to its effects or when expressed excessively within Ewing sarcoma or permissive progenitor cells, EWS/FLI1 triggers cell cycle arrest and cell death [9,10]. This has led to the concept that Ewing sarcomas adhere to the “Goldilocks” principle, wherein EWS/FLI1 must be maintained at precisely balanced levels: an excess of the fusion protein proves toxic, while insufficient levels fail to sustain malignant properties.

Early insights into the influence of fusion expression levels on cell plasticity and tumor behavior emerged from studies involving EWS/FLI1 knockdown. Lowering the expression of EWS/FLI1 leads to alterations in the cytoskeleton of Ewing sarcoma cells, resulting in a larger and more spindle-like morphology, coupled with changes in adhesive properties [11,12,13,14,15]. Subsequent research has identified the regulation of YAP signaling as a downstream mediator of these cytoskeletal modifications [16,17]. Moreover, cells with reduced EWS/FLI1 exhibit heightened activation of the Rho pathway and increased expression of mesenchymal-identity genes [11,13,14,16]. These transcriptional and phenotypic shifts are linked to the promotion of cell migration, invasion, and enhanced metastatic capability. In recent years, advances in genomic technologies have enabled the assessment of heterogeneity in endogenous EWS/FLI1 expression and activity within tumors in their natural context, moving beyond reliance on EWS/FLI1 knockdown and overexpression models. These studies have unveiled variations in the expression and transcriptional activity of the fusion gene within tumor cells. Cells exhibit a spectrum of states ranging from less to more mesenchymal, and this diversity is discernible in tumor-derived cell lines, patient-derived xenografts (PDX) both in laboratory cultures and in living organisms, as well as in patient tumor biopsies [11,18,19,20,20]. In line with findings from genetic knockdown investigations, cells with increased EWS/FLI1 activity exhibit signatures associated with proliferation. On the other hand, cells with decreased transcriptional activity upregulate gene signatures linked to mesenchymal traits and exhibit heightened metastatic potential. [11,19,21].

Gaining insight into the significance of these various cell states in terms of local and metastatic progression, treatment outcomes, and recurrence is currently a dynamic and prominent area of research in this field. In this review, we used the term “EWS/FLI1 high” to describe cells in which EWS/FLI1 is both highly expressed and transcriptionally active. Conversely, we used the term “EWS/FLI1 low” to denote cells in which the transcriptional activity of EWS/FLI1 has been somehow inhibited. EWS/FLI1 high cells typically constitute the majority of Ewing sarcoma cells and generally exhibit proliferative characteristics while remaining relatively immobile. Conversely, cells with low EWS/FLI1 expression constitute a minority within the population, showcasing more mesenchymal and migratory characteristics, and they contribute to the promotion of metastasis. These ‘low’ fusion cells can arise due to the reduction of EWS/FLI1 protein levels or by impairing its function as a transcriptional activator and/or repressor. While these distinctions between EWS/FLI1 high and low states are somewhat simplified and may not encompass the full spectrum of fusion protein activity observed in Ewing sarcoma cells, they provide a fundamental framework for this review. Additionally, they offer a basis for future investigations in the field.

## 3. Mechanism of Gene Repression and Activation

The “high” and “low” states of EWS/FLI1 are determined transcriptionally based on the relative expression levels of fusion target genes. A reduction in the fusion results in a decrease in the expression of genes activated by EWS/FLI1 and an increase in the expression of genes that were previously repressed by it. However, it’s important to note that the mechanisms through which the fusion activates and represses target genes are distinct. While there are still significant gaps in our knowledge, it’s clear that the signatures of activation and repression can be separated depending on the presence of additional regulators that influence these distinct processes [21,22]. Here, we provide a summary of the current understanding of how EWS/FLI1 modifies gene transcription.

### 3.1. EWS/FLI1 Mediated Gene Repression

While considerable research has delved into the mechanisms behind EWS/FLI1-dependent gene activation, the processes governing transcriptional repression remain somewhat less elucidated. However, both direct and indirect mechanisms have been discovered. EWS/FLI1 can directly suppress transcription by binding to wild-type ETS family binding sites located in gene promoters and enhancers. These sites typically contain either a single ETS consensus sequence or a restricted number of GGAA repeats. Upon binding to these sites, EWS/FLI1 displaces the more potent wild-type transcriptional activator, leading to the reduced transcription of the gene [2,8,23]. Indirect mechanisms of gene repression driven by EWS/FLI1 involve the direct activation of transcriptional repressors like NKX2-2 [14,24] and the long non-coding RNA EWSAT1. Additionally, the fusion protein facilitates the recruitment of epigenetic proteins and complexes with repressive functions, including Histone Deacetylases (HDACs) [24,25], LSD1, as well as the NuRD complex, to specific target loci. The combined impact of these direct and indirect mechanisms results in the downregulation of numerous genes, many of which are responsible for regulating mesenchymal identity [25,26]. Remarkably, the de-repression of this mesenchymal signature is associated with the acquisition of metastatic properties. It’s important to note that the cell(s) of origin for Ewing sarcoma are not precisely defined.

### 3.2. EWS/FLI1 Mediated Genes Activation via GGAA Enhancer

EWS/FLI1 possesses dual functions, serving as both a transcriptional activator and a repressor. These dual roles are essential for the effective progression of oncogenesis [27]. One of its extensively researched functions in gene activation involves its role as an anomalous transcription factor that reshapes the epigenome through enhancer reprogramming. EWS/FLI1 serves as a pioneer factor, initiating the formation of new active enhancers by enhancing chromatin accessibility, orchestrating the recruitment of histone acetyl/methyltransferases, and establishing long-distance interactions at GGAA microsatellites [25,28]. These GGAA enhancer sites function as distant regulatory elements for specific gene targets that are selectively upregulated in Ewing sarcoma [29]. Numerous GGAA sites bound by EWS/FLI1 typically exhibit epigenetic silence and lack evolutionary conservation, indicating their restricted involvement in normal transcriptional processes [28]. The absence of conserved GGAA elements likely adds to the challenge of creating representative animal models for Ewing sarcoma [30]. Some of the genes activated by EWS/FLI1 are essential for the process of oncogenesis, and a significant number of them promote cell proliferation [31,32]. The EWS/FLI1-activated signature displays heterogeneity due to the presence of GGAA microsatellite polymorphisms in humans, which can be associated with susceptibility to the disease [33,34]. These genetic variations may account for disparities and fluctuations in the expression of specific target genes observed among various Ewing sarcoma cell lines [33,34]. While much of our current knowledge regarding fusion-dependent gene activation focuses on GGAA repeat microsatellites, it’s worth noting that EWS::FLI1 can also activate target gene expression directly by binding to non-GGAA repeat sites, particularly wild-type ETS binding sites that contain shorter GGAA motifs [35]. An independent study approximated that around 25% of EWS::FLI1 binding sites are located within active cis-regulatory regions that do not contain GGAA repeats [28]. Gaining insight into how the presence and distribution of GGAA and non-GGAA repeats shape and modify the EWS/FLI1 activation signature will be a crucial aspect of future research aimed at exploring intratumoral heterogeneity.

## 4. Transcriptional Regulators and therapeutic opportunities

The association between disease and the association type of the seven reviewed genes are depicted in Table 1.

### 4.1. IGF1R

The insulin-like growth factor 1 receptor (IGF1R) plays a significant role in Ewing sarcoma, contributing to tumor growth, survival, and metastasis [52,53]. The mechanism of action of IGF1R in Ewing sarcoma involves the activation of signaling pathways that promote cell proliferation, inhibit apoptosis (programmed cell death), and enhance tumor cell motility [54]. IGF1R is a receptor tyrosine kinase that becomes active upon binding to its ligands, such as insulin-like growth factor 1 (IGF-1), and insulin-like growth factor 2 (IGF-2) [55]. These ligands are secreted proteins that are present in the extracellular environment surrounding Ewing sarcoma cells. Upon ligand binding, IGF1R undergoes a conformational change and forms a receptor dimer, where two receptor subunits come together and activate each other through transphosphorylation [56,57]. This leads to the activation of the receptor’s intrinsic tyrosine kinase activity. Activated IGF1R phosphorylates specific tyrosine residues within its intracellular domain, creating docking sites for downstream signaling molecules. This triggers the recruitment and activation of various intracellular signaling pathways, including the PI3K-Akt (protein kinase B) [58] pathway and the Ras-Raf-MAPK (mitogen-activated protein kinase) [59,60] pathway. Activation of the PI3K-Akt pathway promotes cell survival and growth by inhibiting apoptosis and stimulating cell cycle progression [61] (Figure 1). Akt, a downstream effector of PI3K, promotes cell survival by inhibiting pro-apoptotic proteins and activating anti-apoptotic proteins [62]. Activation of the Ras-Raf-MAPK pathway leads to increased cell proliferation, migration, and invasion. This pathway triggers a cascade of phosphorylation events, ultimately resulting in the activation of transcription factors involved in cell proliferation [63].

Notably, targeting the IGF-I receptor (IGF-IR) has shown activity against various tumors, but it was notably effective only in a subset of Ewing’s sarcoma patients. In an independent study, researchers compared the gene expression profiles of cells that had developed resistance to three different anti-IGF-IR drugs: human antibodies AVE1642 and Figitumumab, as well as the tyrosine kinase inhibitor NVP-AEW541 [64]. This analysis aimed to identify shared and unique mechanisms of resistance. Among the shared mechanisms, researchers identified two molecular signatures that could differentiate between sensitive and resistant cells. Annotation analysis revealed several pathways commonly affected, including insulin signaling, the MAPK pathway, endocytosis, and modulation of certain members of the interferon-induced transmembrane protein family. In terms of distinctive pathways and processes, resistance to human antibodies primarily involves genes related to neural differentiation and angiogenesis. On the other hand, resistance to NVP-AEW541 is predominantly associated with changes in genes related to inflammation and antigen presentation. Analysis of the pathways commonly altered suggested that resistant cells appear to preserve the IGF-IR internalization and degradation route seen in sensitive cells but consistently reduce their expression. In resistant cells, the absence of the proliferative stimulus, typically maintained by the IGF-I/IGF-IR autocrine loop in Ewing’s sarcoma cells, is counterbalanced by the transcriptional upregulation of IGF-II and insulin receptor-A. This signaling shift appears to promote the MAPK pathway over the v-akt murine thymoma viral oncogene homolog 1 pathway [65,66].

In summary, the intricate nature of the IGF system underscores the need for a thorough analysis of its components to identify patients who can truly benefit from this targeted therapy. Additionally, this supports the notion of simultaneously targeting IGF-IR and insulin receptor-A to enhance treatment efficacy.

### 4.2. DAX-1 (NR0B1)

DAX-1, officially known as NR0B1 (Nuclear Receptor Subfamily 0, Group B, Member 1), is a member of the nuclear receptor superfamily [67]. Nuclear receptors are a class of transcription factors that become active upon binding with small ligands like retinoic acid or steroids [68]. Interestingly, recent computational research has identified flavonoids as potential inhibitors of DAX-1 [5]. Germline mutations in the DAX-1 gene are responsible for two conditions: dosage-sensitive sex reversal (DSS) which occurs in XY individuals, and another condition called adrenal hypoplasia congenita (AHC) is characterized by adrenal insufficiency and hypogonadotropic hypogonadism in males [69,70]. DAX-1 functions as a central regulator of steroidogenesis and plays a role in inhibiting the activity of steroidogenic factor 1 (SF1), a key transcriptional activator responsible for regulating genes involved in the production of steroid hormones [71,72]. Additionally, DAX-1 plays vital roles in various biological processes, including osteoblast differentiation, ion homeostasis and transport, lipid transport, and skeletal development [73,74]. It also contributes to the maintenance of pluripotency in mouse embryonic stem cells by regulating stem cell genes such as Oct-3/4 [74]. Surprisingly, despite its known role in steroidogenesis, DAX-1 has been associated with Ewing sarcoma, a tumor type unrelated to steroidogenic tissues [75,76]. Gene expression studies conducted in two different cell models that ectopically expressed EWS/FLI1 (HEK293 and HeLa cells) revealed that DAX-1 was specifically induced by EWS/FLI1, but not by wildtype FLI1 [77]. Furthermore, DAX-1 was found to be highly expressed in Ewing sarcoma cell lines and tumors, in contrast to its absence in other pediatric tumors like rhabdomyosarcoma or neuroblastoma [77,78]. Notably, DAX-1 expression was shown to depend on EWS/FLI1 expression in the A673 Ewing sarcoma cell line when EWS/FLI1 was knocked down [79]. Another independent study confirmed these findings, solidifying DAX-1 as a target of the EWS/FLI1 oncoprotein [80]. Several functional studies have underscored the critical role of DAX-1 in Ewing sarcoma pathogenesis. Knocking down DAX-1 hampers Ewing sarcoma cell proliferation, induces G1 cell arrest, inhibits the growth of colonies in soft agar, and significantly curtails the growth of xeno-transplanted tumor cells in immunodeficient mice [81].

The consistency of these findings is striking, as they have been replicated in independent laboratories using various Ewing sarcoma cell lines (including TC71, EWS502, and A673) and diverse gene knockdown techniques such as transient retrovirus infection and the inducible expression of EWS/FLI1 shRNAs [82]. An intriguing aspect is the characterization of the gene expression profile regulated by DAX-1 in Ewing sarcoma cell lines, which has shed light on the function of DAX-1 in this context [38]. These investigations have demonstrated that a considerable portion of the genes controlled by EWS/FLI1 in Ewing sarcoma cells also fall under the influence of DAX-1. This underscores the significance of DAX-1 in the pathogenesis of Ewing sarcoma. This implies that DAX-1 not only plays a role in shaping the EWS/FLI1 transcriptional signature but also suggests the existence of a hierarchical control mechanism governed by EWS/FLI1, where specific genes, such as DAX-1, may assume a more prominent position. Investigations into how EWS/FLI1 boosts DAX-1 expression in Ewing sarcoma cells have unveiled an unexpected finding. EWS/FLI1 directly engages with the DAX-1 promoter by binding to a sequence rich in GGAA repeats [79,83]. Remarkably, this motif is a variable microsatellite located within the DAX-1 promoter. It has been confirmed that EWS/FLI1 binds to similar sequences found in the promoters of other genes targeted by EWS/FLI1. This suggests that EWS/FLI1 frequently utilizes this mechanism of gene transcriptional activation to regulate the expression of specific oncogenic genes. This includes genes like Caveolin-1 (CAV1) [84], glutathione S-transferase M4 (GSTM4) [85], FCGRT (Fc fragment of IgG, receptor, transporter, alpha), FVT1/KDSR (3-ketodihydrosphingosine reductase) or ABHD6 (Abhydrolase Domain-Containing Protein) [86], highlighting the broader implications of this regulatory mechanism.

The revelation that DAX-1 expression is governed by a polymorphic GGAA motif repeat prompted an examination. This examination sought to determine whether the count of these repeats might be associated with the level of DAX-1 expression and, consequently, the malignant characteristics of Ewing sarcoma. Several biochemical investigations have indeed established a correlation between the number of GGAA repeats and the degree of promoter activation. These studies indicated that a minimum of nine repeats was required to trigger a response to EWS/FLI1 [87]. However, efforts to establish a clear-cut connection between the length of the microsatellite within the DAX-1 promoter and clinical prognosis have generated inconsistent findings. For instance, it was noted that GGAA microsatellites were longer in African populations, where Ewing sarcoma has a lower incidence but worse overall survival rates compared to European populations [8,87]. Conversely, another study involving 112 patients found that the length of the DAX-1 microsatellite had no discernible influence on clinical outcomes [88]. These discrepancies highlight the complexity of the factors contributing to the clinical behavior of Ewing sarcoma and the need for further research to fully understand the role of DAX-1 in the context of this disease.

Considering all of these findings, DAX-1 emerges as one of the most significant gene targets of EWS/FLI1. The pivotal role of DAX-1 expression in EWS/FLI1-mediated oncogenesis suggests that targeting DAX-1 could be a compelling therapeutic strategy for Ewing sarcoma. In theory, a deeper comprehension of the functions exerted by DAX-1 in Ewing sarcoma and the underlying molecular mechanisms could provide valuable insights into how to disrupt its expression or function within this cancer context [80,89]. DAX-1 is predominantly situated within the nucleus of Ewing sarcoma cells, where it presumably engages with other transcription factors and cofactors to oversee downstream target genes vital for oncogenesis (Figure 2). Interestingly, a fusion of biochemical experiments and gene expression profiling has unveiled a direct interaction between EWS/FLI1 and DAX-1. Notably, both the amino- and carboxyl-termini of DAX-1 were observed to interact with EWS/FLI1 [83,90]. This discovery raises the enticing possibility that disrupting the EWS/FLI1-DAX-1 interaction could open up new therapeutic avenues for Ewing sarcoma treatment. To advance along this research trajectory, it becomes imperative to meticulously map the regions responsible for this interaction. This mapping is essential for the design of small molecules with the potential to disrupt it. Given that the interaction between DAX-1 and EWS/FLI1 could be a crucial component of EWS/FLI1-mediated oncogenesis, interfering with it could hold significant therapeutic promise. DAX-1 has been observed to engage in interactions with various transcriptional regulators in different cellular contexts, primarily with corepressors. For instance, DAX-1 forms an interaction with the Alien corepressor via its silencing domain, and this specific interaction has been demonstrated to play a crucial role in the development of adrenal hypoplasia congenita (AHC) [91]. Additionally, DAX-1 directly interacts with the androgen receptor, NR3C4, inhibiting its activation [92] and it also interacts with other partners such as NR5A1 [93] and ESRRγ [94]. Therefore, conducting experiments that focus on identifying and characterizing these interactions could offer valuable insights into the development of synthetic drugs aimed at targeting these interactions for therapeutic purposes.

One potential advantage of pursuing therapeutic strategies targeting DAX-1 is its restricted expression in specific tissues, primarily the adrenal gland and testis. Consequently, interventions aimed at DAX-1 may predominantly impact these organs. To sum up, research endeavors focused on unraveling the structural aspects of DAX-1 and its mechanisms of interaction with other transcriptional (co)factors are essential. Additionally, the identification of additional protein-protein interactions in the context of Ewing sarcoma could offer fresh perspectives and avenues for the development of novel therapeutic agents (Figure 2).

### 4.3. NKX2.2 (NK2 Homeobox 2)

In Ewing sarcoma, the mechanism of action of NKX2-2 is distinct from its normal developmental role. Instead of acting as a transcriptional activator, NKX2-2 functions as an oncogenic driver in Ewing sarcoma by repressing the expression of genes involved in cell differentiation and promoting a stem-like state [21]. NKX2-2 is upregulated in Ewing sarcoma due to the presence of the EWS/FLI1 fusion protein. In normal development, NKX2-2 is involved in the differentiation of neuronal cells [68]. However, in Ewing sarcoma, the overexpression of NKX2-2 leads to the repression of genes associated with cellular differentiation [95]. NKX2-2 directly binds to the regulatory regions of target genes involved in differentiation, such as NR0B1 (also known as DAX1), and represses their expression [27]. NR0B1 suppression by NKX2-2 contributes to the maintenance of an undifferentiated state in Ewing sarcoma cells.

NKX2-2 plays a role in maintaining a stem-like state in Ewing sarcoma cells. Stem-like cells have the ability to self-renew and differentiate into different cell types. By repressing differentiation-associated genes, NKX2-2 helps to sustain the undifferentiated and proliferative state of Ewing sarcoma cells. In a subsequent investigation, it was discovered that ZEB2 exerted opposite regulation on the target genes of NKX2-2, which significantly overlap with those of EWS/FLI1. NKX2-2, a transcription factor involved in neurodevelopment, is itself a target gene of EWS/FLI1, and it governs a substantial portion of both the gene signature activated and repressed by EWS/FLI1 [14,24]. Indeed, when NKX2-2 is knocked down, it partially mimics the transcriptional state associated with low levels of EWS/FLI1 [14,24,29,82]. Both NR0B1 and NKX2-2 exert inhibition on the expression of a specific group of genes repressed by the fusion by directly binding to their promoters and recruiting Histone Deacetylases (HDACs). In this scenario, NKX2-2, activated by EWS/FLI1, acts as a transcriptional repressor and carries out the fusion’s repressive functions by enlisting chromatin complexes that contain repressive HDACs.

Demonstrated by research conducted by Stephen Lessnick’s team, NKX2-2 is another gene that experiences significant upregulation due to the influence of EWSR1-FLI1 [14,24,96]. NKX2-2 functions as a transcription factor containing a homeodomain and plays pivotal roles in various developmental contexts [97]. In 2006, NKX2-2 was identified as a gene regulated by EWS/FLI1 and was found to be crucial for the oncogenic transformation in Ewing sarcoma [96]. In this particular study, endogenous EWSR1-FLI1 was suppressed using RNA interference, followed by the expression of exogenous EWSR1-FLI1. This experiment led to the identification of NKX2-2 as a gene that is upregulated by EWSR1-FLI1. Functional analysis demonstrated that the suppression of NKX2-2 hindered oncogenic transformation in soft agar assays and impeded tumor development in a xenograft model of Ewing sarcoma. Subsequent investigations revealed that genes repressed by NKX2-2 are enriched in the dataset of genes repressed by EWSR1-FLI1, contributing to mesenchymal differentiation and terminal differentiation processes [14,24].

More recent studies have unveiled that NKX2-2, in conjunction with EWSR1-FLI1, binds to the promoter region of STEAP1 (six transmembrane epithelial antigen of the prostate 1), a gene crucial for the survival of Ewing sarcoma [98]. These finding sheds light on how NKX2-2 cooperates with EWSR1-FLI1 to regulate the expression of STEAP1, although this study did not specifically emphasize direct chromatin-level interactions between NKX2-2 and EWSR1-FLI1 on a genome-wide scale. Due to its distinctive expression pattern in Ewing tumors, several studies have indicated that NKX2-2 serves as a valuable immunohistochemical marker for the detection of Ewing sarcoma. It exhibits higher sensitivity and specificity compared to other genes induced by EWSR1-FLI1, such as NR0B1, and EZH2 [99,100,101].

### 4.4. GLI1

GLI1, short for Glioma-Associated Oncogene Homolog 1, is a transcription factor that falls under the Kruper family of zinc finger proteins [102]. It plays a crucial role within the canonical Hedgehog pathway: when the extracellular Sonic Hedgehog (Shh) molecule binds to the PTCH receptor, this event initiates the release of Smooth (SMO) from the PTCH-SMO complex [103]. Subsequently, activated SMO liberates GLI1 from its complex with Suppressor of Fused (SUFU), allowing GLI1 to translocate into the nucleus. Inside the nucleus, GLI1 takes charge of regulating the transcription of genes associated with normal cell growth and differentiation, including those involved in embryonic pattern formation [104,105]. While this pathway is primarily active during embryonic development, it continues to operate in some adult tissues, where it plays a role in maintaining tissue homeostasis and stem-cell functions [106,107]. An intriguing connection between EWS/FLI1 and GLI1 in Ewing sarcoma cells was described by Zwerner et al. Their research demonstrated that NIH3T3 cells expressing EWS/FLI1 exhibited the expected malignant characteristics, alongside an increase in GLI1 levels [108].

Furthermore, experiments involving RNA interference demonstrated that reducing GLI1 expression led to a decrease in the transformed phenotype, as indicated by a reduction in anchorage-independent growth. This finding underscores the significant role that GLI1 plays in maintaining the malignant characteristics induced by EWS/FLI1. Interestingly, when SUFU was overexpressed—expected to inhibit GLI1—it produced similar effects in NIH3T3 cells. In TC32 Ewing sarcoma cells, the knockdown of EWS/FLI1 using RNA interference resulted in a reduction in GLI1 expression levels. Additionally, chromatin immunoprecipitation (ChIP) studies provided evidence that GLI1 is a direct target of EWS/FLI1 [109]. Moreover, when a short hairpin RNA (shRNA) designed to target GLI1 was applied to the Ewing sarcoma cell line TC32, it led to the inhibition of the transformed phenotype. This inhibition was evident through a reduction in anchorage-independent growth [108]. Interestingly, in contrast to the usual findings in other cancer types, the abnormal regulation of GLI1 in Ewing sarcoma occurs independently of Sonic Hedgehog (Shh) signaling. Activation of Shh did not result in observable phenotypic changes, nor did the pharmacological inhibition of SMO using cyclopamine, an inhibitor of Shh signaling through direct SMO binding. Subsequently, Joo et al. [110] confirmed that Ewing primary tumors exhibited high levels of GLI1 expression. Additionally, these investigators employed RNA interference to establish that GLI1 expression in Ewing sarcoma cells (specifically TC71) relies on EWS/FLI1, and it was confirmed that GLI1 expression plays a crucial role in maintaining the transformed phenotype.

Surprisingly, upon revising gene expression profiles, it was uncovered that genes traditionally thought to be controlled by EWS/FLI1, such as NKX2-2, Patched (PTCH), or GAS1, were, in fact, dependent on GLI1 expression. This discovery implies that the gene expression network regulated by EWS/FLI1 functions within a hierarchical structure where GLI1 assumes a prominent position. Disruption of the Sonic Hedgehog (Shh)–GLI1 pathway has been shown to result in tumorigenesis and the emergence of aggressive traits, including disease progression, metastasis, and therapy resistance, in various types of cancer. These cancers encompass a spectrum of malignancies, including basal cell carcinomas, colorectal carcinoma, breast cancer, as well as bone and soft tissue sarcomas [31,110]. Considering the significance of the Shh–GLI1 pathway in cancer, various therapeutic approaches have been developed over the years to block this pathway. One of these strategies involved the search for small molecule inhibitors targeting the Shh–GLI1 pathway. As a result, inhibitors of the Shh–GLI1 pathway, such as cyclopamine, have been effectively tested in several cancer types, including medulloblastoma, small-cell lung cancer (SCLC), pancreatic adenocarcinoma, gastric adenocarcinomas, and esophageal cancer [111,112]. In this pathway, Sonic Hedgehog (Shh) binding to the PTCH receptor triggers the release of Smooth (SMO) from the PTCH-SMO complex. Subsequently, activated SMO liberates GLI1 from its complex with Suppressor of Fused (SUFU), allowing GLI1 to translocate into the nucleus, where it orchestrates the transcription of genes crucial for normal cell growth, differentiation, and embryonic pattern formation [112,113,114]. This association between EWS/FLI1 and GLI1 in Ewing sarcoma cells was initially demonstrated by Zwerner et al., who observed that NIH3T3 cells expressing EWS/FLI1 exhibited the expected malignant characteristics alongside elevated levels of GLI1 [108].

Furthermore, when GLI1 expression was specifically reduced using RNA interference, a noticeable decrease in the transformed phenotype was observed, as evidenced by a reduction in anchorage-independent growth. This finding underscores the pivotal role played by GLI1 in sustaining the malignant phenotype induced by EWS/FLI1. Interestingly, when SUFU, which is expected to inhibit GLI1, was overexpressed, similar effects were observed in NIH3T3 cells. In TC32 Ewing sarcoma cells, the silencing of EWS/FLI1 through RNA interference led to a corresponding decrease in GLI1 expression levels. Chromatin immunoprecipitation (ChIP) studies further provided evidence that GLI1 is indeed a direct target of EWS/FLI1 [23,109,115]. Additionally, when a short hairpin RNA (shRNA) targeting GLI1 was employed in the Ewing sarcoma cell line TC32, it resulted in the inhibition of the transformed phenotype, as evidenced by a reduction in anchorage-independent growth. Intriguingly, and in contrast to the usual observations in other cancer types, the dysregulation of GLI1 in Ewing sarcoma appears to be independent of Sonic Hedgehog (Shh) signaling. Activation of Shh did not induce phenotypic changes, nor did pharmacological blockade of SMO using cyclopamine, which is an inhibitor of Shh signaling through direct binding to SMO [108,109].

Disruption of the Shh–GLI1 pathway has been consistently associated with tumorigenesis and the emergence of aggressive phenotypes, including disease progression, metastasis, and resistance to therapy, in various cancer types. These encompass basal cell carcinomas, colorectal carcinoma, breast cancer, as well as bone and soft tissue sarcomas [104,116,117]. Recognizing the pivotal role of the Shh–GLI1 pathway in cancer, therapeutic strategies aimed at blocking this pathway have been developed over the years. One notable approach has involved the search for small molecule inhibitors targeting the Shh–GLI1 pathway. Consequently, inhibitors of this pathway, such as cyclopamine, have shown promising results in certain cancer types, including medulloblastoma [117,118,119], pancreatic adenocarcinoma [119,120], small-cell lung cancer (SCLC) [119,121], gastric adenocarcinomas [122], and esophageal cancer [119,123]. However, there remains an urgent need for further functional studies to precisely elucidate the role of this pathway in the development and progression of Ewing sarcoma. Such investigations could potentially lead to the development of new compounds or small molecules with enhanced efficacy for targeting GLI1, either as standalone treatments or in combination with conventional chemotherapeutic approaches (Figure 3).

### 4.5. EZH2 (Enhancer of Zeste)

As a direct downstream target of EWSR1-FLI1, EZH2 plays a critical role in preserving the stemness characteristics of Ewing sarcoma cells [124,125,126,127]. EZH2 is an integral component of the Polycomb repressor complex 2 (PRC2) and exerts its function by methylating lysine 27 on histone 3 (H3K27), thereby repressing gene expression [128]. The functions of EZH2 extend to stem cell maintenance, embryonic development, and cell differentiation [129]. Aberrantly elevated levels of EZH2 expression in cancer have been linked to unfavorable clinical outcomes [130,131].

In Ewing sarcoma, EWSR1-FLI1 binds to the EZH2 promoter, prompting the transcriptional upregulation of EZH2. When gene expression profiling was conducted in primary Ewing sarcoma tumors and stem cells expressing EWSR1-FLI1. it was discovered that HOX genes, acting downstream of EZH2, disrupt developmental transcription programs, thereby conferring the stemness traits to Ewing sarcoma tumors [132]. The inhibition of EZH2 in Ewing sarcoma cells resulted in a loss of their ability to grow independently in vitro and their tumorigenic potential in vivo, highlighting EZH2 as a potential therapeutic target. Mechanistically, EZH2 inhibition led to the increased expression of genes associated with neuroectodermal and endothelial differentiation, such as EMP1, EPHB2, GFAP, and GAP43 [126]. In summary, downstream of EWSR1-FLI1, EZH2 acts to hinder the differentiation of Ewing sarcoma cells and sustains their stem cell-like state, which contributes to oncogenic transformation and the progression of tumors.

### 4.6. Forkhead Box (FOX) of Transcription Factors

Forkhead box proteins comprise a large family of transcriptional regulators that possess a shared DNA binding domain known as the forkhead domain [133]. These 19 subgroups, designated from FOXA to FOXS, are categorized based on both sequence homology within and outside the forkhead domain [134]. FOX proteins oversee gene networks that play roles in cell cycle proliferation, metabolic processes, progression, differentiation, senescence, apoptosis, and survival [135]. Given their involvement in various critical cellular processes, it comes as no surprise that these transcription factors have demonstrated roles in cancer. Intriguingly, certain members of this family have been found to function as tumor suppressor genes, whereas others have been identified as pro-oncogenic. Examples of both these contrasting functions have emerged in the context of Ewing sarcoma. The FOXO subgroup, which includes FOXO1, FOXO3A, FOXO4, and FOXO6, play crucial roles as negative regulators of both cell proliferation and survival [136,137,138]. These proteins promote cell cycle arrest at the G1 phase, initiate apoptosis, and facilitate DNA repair processes [139,140,141]. Hence, they are recognized as genuine tumor suppressors. As an illustration, in prostate cancer, FOXO1 is often observed to be downregulated at the transcriptional level. Elevating its expression in prostate cancer cells has been shown to impede both cell proliferation and survival [142,143]. Moreover, FOXO1 has also been demonstrated to oversee other critical aspects of cancer, including the regulation of angiogenesis [144]. As a result, the loss of FOXO1 function leads to heightened blood vessel formation and encourages both endothelial cell migration and proliferation [145].

The transcriptional activity of FOXOs is controlled by alterations in their cellular localization, a process mediated by protein kinases like serum/glucocorticoid kinase (SGK) and protein kinase B (AKT) [146]. These transcription factors can also experience various post-translational modifications that govern their activity, such as deacetylation facilitated by Sirt1 and ubiquitination facilitated by Skp2 and Mdm2 [137,147]. In Ewing sarcoma cells, EWS/FLI1 binds to the promoter of FOXO1 and suppresses its expression [148]. In accordance with this, primary Ewing sarcoma exhibits lower levels of FOXO1 expression compared to other tissues. When FOXO1 was induced in two Ewing sarcoma cell lines, A673 and SKNMC, it led to reduced cell proliferation and diminished capability to form colonies in soft agar. These findings confirm that FOXO1 functions as a tumor suppressor in Ewing sarcoma and that its inhibition is critical for the growth of this cancer [149]. Remarkably, EWS/FLI1 indirectly influences the subcellular localization of FOXO1, thereby modulating its transcriptional activity [150]. CDK2, which is elevated by EWS/FLI1 and functions as a suppressor of FOXO1’s transcriptional activity, along with AKT-mediated phosphorylation of FOXO1, work together to hinder its transportation into the nucleus, ultimately inhibiting its transcriptional function [151,152]. These discoveries highlight that EWS/FLI1 inhibits FOXO1 activity through multiple mechanisms within Ewing sarcoma cells. Given FOXO1’s role as a tumor suppressor in Ewing sarcoma, a promising therapeutic strategy could involve the reactivation of FOXO1. In this context, methylseleninic acid (MSA), a chemical compound previously demonstrated to reactivate FOXO1 in prostate cancer, was evaluated in Ewing sarcoma cells [153]. The treatment of Ewing sarcoma cells with MSA resulted in the induction of FOXO1 expression in a concentration-dependent manner, and this correlated with cell death through apoptosis. To some extent, this effect was attributed to FOXO1, as the reduction of endogenously induced FOXO1 significantly diminished the apoptotic impact of MSA. Importantly, in an orthotopic mouse xenotransplantation model, the administration of MSA notably reduced tumor growth, indicating that MSA holds promise as a potential therapeutic approach in Ewing sarcoma [154]. Nevertheless, it’s crucial to consider that elevated levels of selenium are typically linked to toxicity, which can pose challenges in this approach. Therefore, any potential utilization of MSA should involve the use of effective yet low doses, in combination with conventional chemotherapeutic medications, to achieve the desired anti-tumor effects. Specifically, MSA has already demonstrated synergistic effects when combined with certain chemotherapeutic drugs commonly employed in Ewing sarcoma treatment, such as etoposide or doxorubicin [155] (Figure 4). As the reactivation of FOXO1 has demonstrated effectiveness in Ewing sarcoma cells in both laboratory settings (in vitro) and living organisms (in vivo), further research is imperative to unravel the intricate mechanisms governing FOXO1 expression and its transcriptional activity. This quest for understanding is essential for pinpointing novel therapeutic targets.

FOXM1 belongs to the FOX family of transcription factors but, in contrast to FOXO, it plays a pro-oncogenic role in cancer [156]. In reality, FOXM1 is among the most frequently upregulated genes observed in solid tumors [157]. At first, FOXM1 was characterized as a mammalian transcription factor primarily associated with cell proliferation, being expressed in actively dividing cells while absent in quiescent or fully differentiated cells. Furthermore, over time, FOXM1 has been implicated in various other processes including cell migration, invasion, angiogenesis, metastasis, and responses to oxidative stress [158,159,160]. Christensen et al. demonstrated that EWS/FLI1 increased the expression levels of FOXM1 in four different Ewing sarcoma cell lines. However, it was observed that the mechanism behind this upregulation appeared to be indirect [51]. In line with this, FOXM1 is found to be expressed at elevated levels in both Ewing sarcoma cell lines and primary tumors. To assess the significance of FOXM1 in Ewing sarcoma pathogenesis, the researchers conducted FOXM1 knockdown experiments, revealing a significant decrease in anchorage-independent growth upon FOXM1 downregulation. Interestingly, they also explored pharmacological approaches aimed at reducing FOXM1 levels in Ewing sarcoma cells, yielding noteworthy results. For instance, the thiazole antibiotic Thiostrepton has been demonstrated to function as a proteasomal inhibitor [161]. Additionally, Thiostrepton was shown to physically interact with FOXM1, thereby hindering FOXM1’s ability to bind to its target promoters [162]. Treatment with Thiostrepton led to the inhibition of FOXM1 expression, which was accompanied by an increase in apoptosis observed in several Ewing sarcoma cell lines [51,154]. Thiostrepton was also demonstrated to effectively inhibit tumor growth in mouse xenograft models [163]. Remarkably, in this study, Thiostrepton was capable of simultaneously suppressing the expression of EWS/FLI1 at both the mRNA and protein levels in three different Ewing cell lines and tumors originating from mouse xenograft models treated with Thiostrepton [163]. While the specific mechanism by which Thiostrepton induces the down-regulation of EWS/FLI1 was not investigated, these findings suggest that this drug may exhibit higher effectiveness in Ewing sarcoma tumors compared to other types of tumors. As previously mentioned, FOXM1 is commonly overexpressed in various cancers and plays a role in each hallmark of cancer. Therefore, there is a proposition that targeting FOXM1 could present a promising approach for cancer treatment. Some have even suggested that FOXM1 might be the ‘’Achilles heel’’ of cancer [164]. In summary, these results collectively indicate that targeting FOXM1 could also be a potential avenue for Ewing sarcoma treatment (Figure 5).

## 5. Discussion

Genes that are activated transcriptionally by GGAA microsatellites represent a tumor-specific mechanism, offering the potential for targeted therapies in Ewing sarcoma treatment. Additionally, proteins that interact with EWSR1-FLI1 can be candidates for specific therapies in Ewing sarcoma treatment. To develop novel therapies, an improved understanding of the molecular biology of Ewing sarcoma is imperative. Significant efforts have been dedicated to identifying the transcriptional targets of EWSR1-FLI1, with a substantial number of genes identified as crucial for EWSR1-FLI1-mediated tumorigenesis. However, despite extensive research, only a few targets have been clinically proven to have prognostic or therapeutic significance. Ewing sarcoma cells exhibit notable heterogeneity, transitioning between functionally distinct states depending on fluctuations in EWSR1-FLI1 expression. Cells with high EWSR1-FLI1 expression levels demonstrate exponential proliferation, while those with low EWSR1-FLI1 expression tend to exhibit migratory and invasive characteristics. [136].

The versatility and ability of Ewing sarcoma cells to adapt to their surroundings are supported by their capacity to modify their gene expression and metabolic patterns [137,138]. Hence, there is an immediate necessity for additional research aimed at gaining a comprehensive understanding of the dynamics and clinical consequences associated with the development and advancement of Ewing sarcoma. Such investigations have the potential to uncover novel therapeutics that specifically target EWSR1-FLI1-mediated epigenetic and transcriptional signaling. These therapies may offer enhanced efficacy, either as standalone treatments or in combination with conventional chemotherapeutic approaches. Since 2007, multiple studies have been carried out employing high-throughput chemical, RNAi, and CRISPR-Cas9 screens to identify therapeutic targets in an unbiased manner [61,126,127,128,129,130,131,132,133,134,135,139,140]. These screens are also being utilized to identify genes involved in drug resistance and to discover synergistic drug regimens that are more potent than equally effective doses of its components. Successful combination therapies might allow for reduced dosing, leading to more manageable side effects. Collectively, studies on the transcriptional and epigenetic alterations induced by EWSR1-FLI1 have provided valuable insights into the initiation and development of Ewing sarcoma. Furthermore, their implementation via chemical and genetic screens has unveiled exciting opportunities for future Ewing sarcoma treatments. This is especially beneficial for treating tumors in children, who often have a lower tolerance for adverse side effects. High-throughput chemical and genetic screens have emerged as a noteworthy area of focus in Ewing sarcoma research.

## 6. Conclusions

This review primarily focuses on the downstream regulatory network of EWS/FLI1, with a specific emphasis on the target genes that are either up-regulated or down-regulated by EWS/FLI1. The main objective is to underscore the potential clinical significance of studying these genes and their role in developing targeted therapies for Ewing sarcoma. Additionally, we have identified several current unresolved questions related to the pathways and underlying mechanisms responsible for the functional effects of these genes in this disease. These unanswered issues hold great promise in providing valuable insights for future research in understanding Ewing sarcoma. There are still numerous aspects regarding EWS/FLI1 target genes that remain puzzling, and acquiring a more comprehensive understanding of these mechanisms could pave the way for designing more precise and less detrimental treatments specifically tailored to combat Ewing sarcoma.

## Figures and Tables

**Figure 1 ijms-24-15173-f001:**
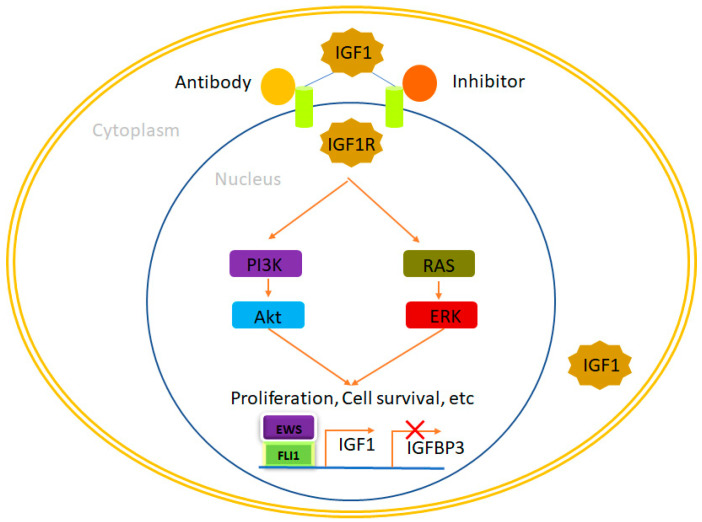
IGF1R, a therapeutic opportunity in Ewing sarcoma. EWS/FLI1 binds to the promoters of its target genes, leading to an elevation in IGF1 expression while simultaneously reducing the expression of IGFBP3. This dynamic interplay results in the increased availability of IGF1, which subsequently interacts with IGF1R and initiates the IGF1 pathway. This pathway plays a pivotal role in regulating cellular processes such as proliferation, survival, and more. Notably, the activation of the IGF1 pathway can be effectively suppressed through the administration of IGF1R inhibitors and antibodies, providing a means to modulate this critical signaling cascade. In this context, IGF1 denotes insulin-like growth factor 1, while IGFBP3 refers to insulin-like growth factor binding protein 3.

**Figure 2 ijms-24-15173-f002:**
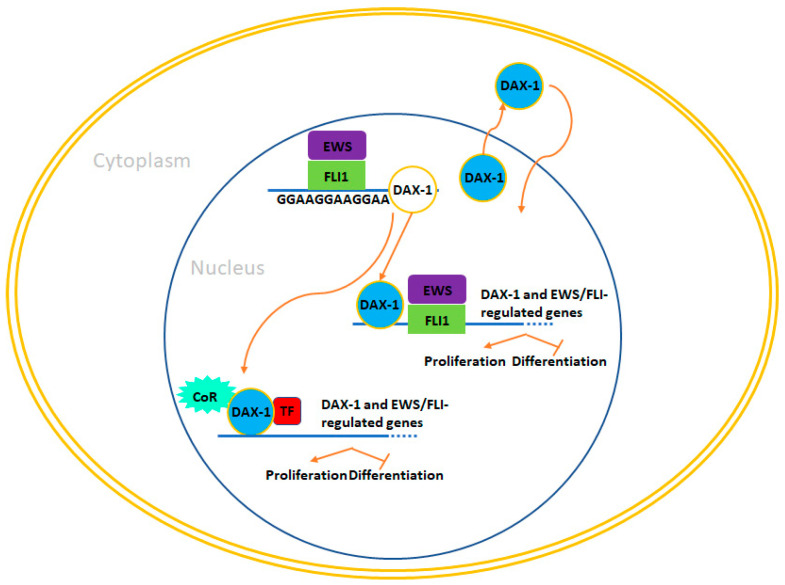
DAX1, a therapeutic opportunity in Ewing sarcoma. EWS/FLI1 plays a significant role in elevating the expression of DAX-1 within Ewing sarcoma cells, primarily achieved through its direct interaction with a polymorphic GGAA microsatellite situated within the DAX-1 promoter region. Since the expression of DAX-1 is vital for the oncogenic effects mediated by EWS/FLI1, it remains crucial to investigate potential interactions with other nuclear transcription factors and/or co-repressors in Ewing sarcoma cells. This exploration could unveil novel avenues for therapeutic development, focusing on molecules designed to disrupt these interactions. Promising therapeutic targets may include compounds that hinder the binding of EWS/FLI1 to the GGAA-rich motifs in the DAX-1 promoter or drugs aimed at disrupting the EWS/FLI1-DAX-1 interaction, potentially vital for regulating the expression of specific genes. Furthermore, the subcellular localization and, consequently, the function of DAX-1 can be potentially modulated by targeting its C-terminal domain when it undergoes alterations.

**Figure 3 ijms-24-15173-f003:**
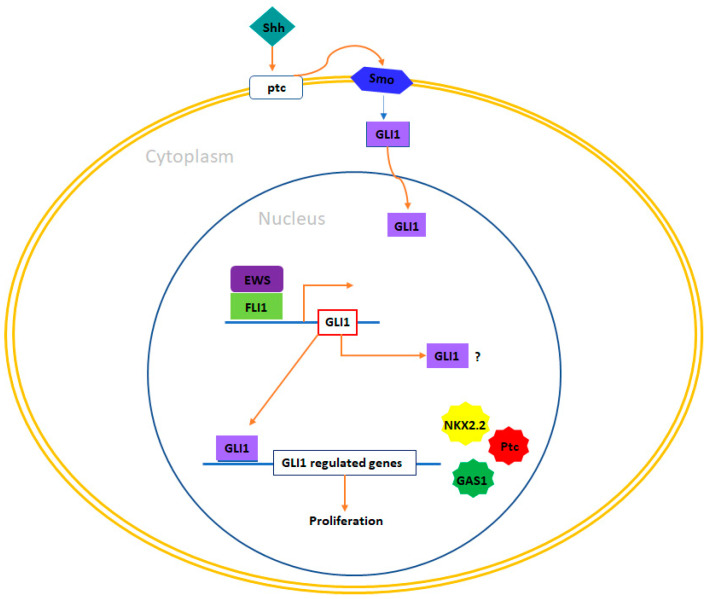
GLI1 and therapeutic opportunities in Ewing sarcoma. In Ewing sarcoma cells, GLI1 stands out as a direct target gene that experiences upregulation induced by EWS/FLI1. Functional investigations have revealed that the expression of GLI1 holds significant relevance in maintaining the transformed phenotype within this system. Additionally, certain genes under the transcriptional influence of EWS/FLI1 are contingent upon GLI1 expression, including NKX2-2, PATCH, and GAS1. Exploring therapeutic prospects in this context could encompass the utilization of compounds capable of inhibiting GLI1-mediated transcription, such as arsenic trioxide (As2O3). Furthermore, it may be valuable to investigate a potential correlation between the pattern of GLI1 expression and prognosis in Ewing sarcoma, considering that such a connection between GLI1 expression and adverse prognosis has been established in other tumor types, such as breast cancer and bone and soft tissue sarcomas.

**Figure 4 ijms-24-15173-f004:**
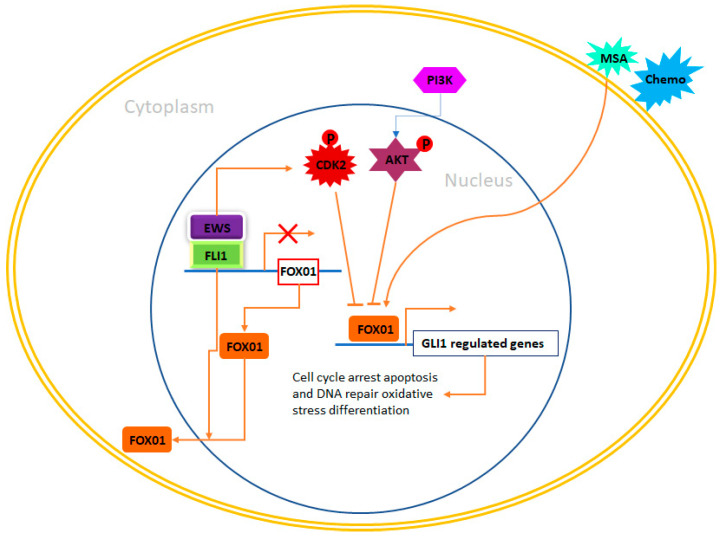
FOXO1 and therapeutic opportunities in Ewing sarcoma. In Ewing sarcoma cells, FOXO1 serves as a direct target gene subject to repression by EWS/FLI1. Notably, functional investigations have underscored the role of FOXO1 as a tumor suppressor within the context of Ewing sarcoma. From a therapeutic standpoint, Methane Sulfonic Acid (MSA) emerges as a potential treatment option, particularly in combination with other chemotherapeutic agents like doxorubicin or etoposide. Nevertheless, the precise mechanism through which MSA operates in the context of Ewing sarcoma remains undisclosed. Additionally, there is a pressing need to elucidate FOXO1 activities that may be modulated by kinases, such as CDK2 and AKT, alongside the regulation of its subcellular localization, with a focus on ascertaining if these processes are influenced by EWS/FLI1.

**Figure 5 ijms-24-15173-f005:**
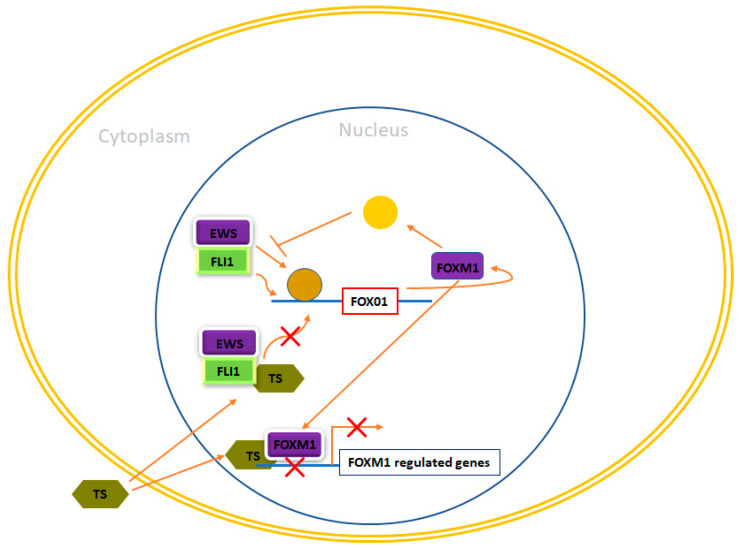
FOXM1 and therapeutic opportunities in Ewing sarcoma. In Ewing sarcoma cells, FOXM1 experiences upregulation due to the influence of EWS/FLI1, although the precise nature of this regulation, whether it is direct or indirect, remains unclear. Thiostrepton (TS) has been observed to impede FOXM1 in Ewing sarcoma cells, leading to a reduction in their neoplastic characteristics. Nonetheless, the specific mechanism responsible for these effects remains enigmatic. Furthermore, FOXM1 has exhibited the potential to inhibit EWS/FLI1, likely through an indirect mechanism that requires further elucidation. Additionally, TS has been demonstrated to inhibit EWS/FLI1 expression in Ewing cells, although the exact mechanism behind this action remains unknown.

**Table 1 ijms-24-15173-t001:** Seven reviewed genes and their association with disease.

Sr No	Genes	Association Type	Disease	References
1	IGF1R	Biomarker, Altered Expression	Ewing sarcoma, Alzheimer’s disease	[36,37]
2	DAX-1	Biomarker	Ewing sarcoma, Neoplasm	[38,39]
3	NKX2.2	Altered expression, Biomarker	Ewing sarcoma, soft tissue neoplasm	[40,41]
4	GLI1	Biomarker	Liver carcinoma, Ewing sarcoma, childhood medulloblastoma	[42,43]
5	EZH2	Altered expression, Biomarker	Liver carcinoma, breast carcinoma, Ewing sarcoma	[44,45,46]
6	FOXO1	Biomarker, Altered expression	Breast carcinoma, Neoplasm, Ewing sarcoma	[47,48,49]
7	FOXM1	Altered expression, Biomarker	Childhood osteosarcoma, Ewing sarcoma	[50,51]

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
