# Peer review of "EWS/FLI1 Characterization, Activation, Repression, Target Genes and Therapeutic Opportunities in Ewing Sarcoma"

_ijms, 2023, doi:10.3390/ijms242015173_

Round 1

Reviewer 1 Report

Yasir and colleagues provide a comprehensive overview of some selected targets of EWS-FLI1, highlighting  novel potential opportunities for treatment in Ewing sarcoma. The review is well organized, the topic is covered and the highlighted molecules are interesting. The figures are attractive and help getting the message across. I only have one minor suggestion in section 4.1 – IGF1R: I suggest to expand this paragraph evidencing for instance the crucial role of an active IGF system in the transforming potential of EWS-FLI1 itself. It should be considered that among the targeted therapies, anti-IGF agents are those with the best response in Ewing sarcoma.

Author Response

Thank you for the careful review of the manuscript.

Regarding the role of IGF1R in Ewing sarcoma, as you recommended, we have provided research findings that establish a clear link between anti-IGF agents and the transforming potential of EWS-FLI1 in Ewing sarcoma.

We appreciate your careful reading and critical comments on our manuscript. We have revised the manuscript based on your comments. Your feedback has been invaluable in helping us to improve the clarity, accuracy, and overall quality of our work.

Reviewer 2 Report

The manuscript read well, I would suggest that

1. the "therapeutic opportunities in Ewing sarcoma" should be one seperate section.

2. the Discussion section was not well orgnized, a little hard to read, pls revise.

pls avoid use long sentence through the whole text.

Author Response

  1. The "therapeutic opportunities in Ewing sarcoma" should be one seperate section

Thank you for carefully reviewing our manuscript and for your constructive feedback.

We appreciate your suggestion to separate the 'therapeutic opportunities in Ewing sarcoma' into a separate section. After careful consideration, we have chosen to maintain the current structure where we discuss transcriptional regulators and therapeutic opportunities. We believe that this approach allows for a more coherent and effective presentation of the information. We understand your concern, but we have reasons to believe that separating these sections may disrupt the flow and coherence of the manuscript. We appreciate your understanding.

  1. the Discussion section was not well organized, a little hard to read, pls revise.

We appreciate your comments regarding the organization of the Discussion section. We understand your concern that it may not be well-organized and may be challenging to read. We aimed to address the research objectives and follow a logical progression of ideas. However, we acknowledge that there is room for improvement in terms of clarity and readability. Based on your recommendation, we have revised the Discussion section to make it more reader-friendly.

  1. pls avoid use long sentence through the whole text.

Based on your recommendation, we have revised the sentences that were too long and difficult to understand clearly into shorter ones to make them easier to read throughout the manuscript.

We appreciate your careful reading and critical comments on our manuscript. We have revised the manuscript based on your comments. Your feedback has been invaluable in helping us to improve the clarity, accuracy, and overall quality of our work.

Reviewer 3 Report

The authors aimed to review the role of the EWS/FLI fusion protein in Ewing sarcoma by exploring its general mechanism of activation and elucidating its implications for tumor heterogeneity. 

the topic is interesting and the review is well written.

I would specify the narrative nature of this review.

A table which resume all pathways and potential targets would be an added value.

Nonetheless, a very interesting narrative review.

Author Response

A table which resume all pathways and potential targets would be an added value.

Thank you for your careful review of the manuscript.

We have added the table in the manuscript as Table 1. We have summarized the potential targets and their associations to the disease.

We appreciate your careful reading and critical comments on our manuscript. We have revised the manuscript based on your comments. Your feedback has been invaluable in helping us to improve the clarity, accuracy, and overall quality of our work.